# Relapsed Rhabdomyosarcoma

**DOI:** 10.3390/jcm10040804

**Published:** 2021-02-17

**Authors:** Christine M. Heske, Leo Mascarenhas

**Affiliations:** 1Pediatric Oncology Branch, National Cancer Institute, National Institutes of Health, Bethesda, MD 20892, USA; 2Cancer and Blood Disease Institute, Children’s Hospital Los Angeles, Division of Hematology/Oncology, Department of Pediatrics and Norris Comprehensive Cancer Center, Keck School of Medicine, University of Southern California, Los Angeles, CA 90027, USA; lmascarenhas@chla.usc.edu

**Keywords:** rhabdomyosarcoma, relapse, risk-factors, prognosis, chemotherapy, clinical trials

## Abstract

Relapsed rhabdomyosarcoma (RMS) represents a significant therapeutic challenge. Nearly one-third of patients diagnosed with localized RMS and over two-thirds of patients with metastatic RMS will experience disease recurrence following primary treatment, generally within three years. Clinical features at diagnosis, including primary site, tumor invasiveness, size, stage, and histology impact likelihood of relapse and prognosis post-relapse. Aspects of initial treatment, including extent of surgical resection, use of radiotherapy, and chemotherapy regimen, are also associated with post-relapse outcomes, as are features of the relapse itself, including time to relapse and extent of disease involvement. Although there is no standard treatment for patients with relapsed RMS, several general principles, including tissue biopsy confirmation of diagnosis, assessment of post-relapse prognosis, determination of the feasibility of additional local control measures, and discussion of patient goals, should all be part of the approach to care. Patients with features suggestive of a favorable prognosis, which include those with botryoid RMS or stage 1 or group I embryonal RMS (ERMS) who have had no prior treatment with cyclophosphamide, have the highest chance of achieving long-term cure when treated with a multiagent chemotherapy regimen at relapse. Unfortunately, patients who do not meet these criteria represent the majority and have poor outcomes when treated with such regimens. For this group, strong consideration should be given for enrollment on a clinical trial.

## 1. Introduction

Rhabdomyosarcoma (RMS) comprises the largest group of soft tissue sarcomas affecting children and adolescents, with approximately 350 new cases diagnosed each year in the United States [1]. Primary therapy consists of a risk-based multidisciplinary approach using multiagent chemotherapy and local control with surgical resection and/or ionizing radiation [2]. While treatment advances over the past several decades have significantly improved the long-term survival outcomes for patients with localized disease, an estimated 25–30% of these patients still suffer relapses; for patients with metastatic disease, relapse rates approach 70% [3]. Following relapse, there is no widely accepted standard of care and outcomes generally remain poor [2]. This article will review the current state of knowledge with regard to the presentation and prognostic factors for relapsed RMS as well as the current and future potential treatment approaches for this group of patients.

## 2. RMS Relapse: A Historical Perspective

Between the 1970s and the early 2000s, results from a number of European RMS trials reported relapse rates of 31% [4], 29% [5], and 36% [6] for patients who presented initially with non-metastatic disease treated on the Italian RMS studies (RMS 79, 88, 96), the Cooperative Weichteilsarkom Studiengruppe (CWS) studies (CWS 81, 86, 91, 96), and the International Society of Paediatric Oncology (SIOP) Malignant Mesenchymal Tumor (MMT) studies (MMT-84, 89, 95), respectively. Results from the North American Intergroup Rhabdomyosarcoma Study (IRS) I, which was conducted during the early part of this period, reported a similar relapse rate of 34% in all newly diagnosed RMS patients, including those with both localized and metastatic disease at diagnosis [7]. Patients treated on IRS III-IV, conducted during the 1980s and 1990s, experienced an improved relapse rate of 26%, compared to those treated on IRS-I [8]. This improvement was primarily the result of fewer relapses in patients with non-metastatic embryonal RMS and was attributed to a decreased frequency of systemic relapse due to the introduction of more effective chemotherapeutics [9].

Outcomes following first relapse on the earliest study were dismal, with only 6% of patients treated on IRS-I achieving long-term disease control post-relapse [7]. For those treated on IRS-III/IV, this number improved to 17% (5-year overall survival (OS)) [8]. Data from these studies suggested that an even greater proportion of patients who relapsed following treatment for primary localized disease were salvageable with post-relapse treatment and had 5-year OS of 24% reported in the Italian series [4], 3-year OS of 37% reported in the SIOP-MMT studies [6], and 5-year OS of 21% reported in the CWS studies [10]. While the risk of relapse and post-relapse death has largely remained unchanged since these data were reported [2,3], these early data were fundamental to the field, as they identified risk factors for subsequent relapse and provided the basis on which to define RMS risk groups going forward.

## 3. Timing and Pattern of Relapse

Across all of these studies, most relapses occurred within a relatively short timeframe, regardless of the regimen or specific patient population. In the SIOP-MMT studies, patients with initially localized disease had a median time to relapse of 14 months (range 2–102 months), with 78% of relapses occurring within 2 years of initial diagnosis [11]. In similar patient populations, the Italian group observed median time to relapse of 17.8 months [4], and the CWS studies reported a median time to relapse of 1.5 years (range 0.2–13.5 years) [5], with 67% of the relapses occurring within 1 year of completion of primary therapy [12]. The results from the IRS III-IV studies, which included patients with both localized and metastatic disease, reported a median time to relapse that was slightly shorter (1.1 year, range 1 week–9 years) with nearly all relapses (95%) detected within 3 years from the start of initial treatment [8].

In the series of patients with localized RMS, the majority of relapses (64–76%) were isolated locoregional relapses [4,5,6,11]. The SIOP-MMT series, which reported the most detailed of these data, indicated that the vast majority (81–87%) of these local relapses occurred exclusively at the primary site, with the remainder occurring in the locoregional nodes or in both sites [6,11]. Metastatic relapses in this population were more likely to occur earlier [4] and in patients who had initially presented with group III (local tumor without distant spread than cannot be completely resected) disease [11]. The most common sites of metastatic relapses included thorax, abdomen, central nervous system, and bone, and most patients with metastatic relapses had multiple synchronous metastatic sites involved [5]. Combined (local and metastatic) relapses were the least likely to occur in this patient group and were detected in approximately 10% of patients [5,6]. Not surprisingly, the IRS III-IV studies, which included patients with both localized and metastatic RMS at diagnosis, reported a higher proportion of patients (41%) that relapsed with distant metastases [8] compared to the studies describing patients with initially localized disease.

For infants with RMS, although a similar pattern of relapse has been reported, higher rates of local failure have been noted in multiple studies, particularly for patients with group III tumors [13,14,15,16,17]. This has largely been attributed to the use of less aggressive local control measures in this population, as opposed to biological differences between tumors in infants and those in older patients [14,16,17].

## 4. Risk Factors for Outcomes Post-Relapse

Retrospective analyses of the patients who had relapsed following treatment on the early RMS studies revealed important insights into the risk factors associated both with relapse risk and the likelihood of survival after recurrence. These include clinical factors present at diagnosis, aspects of upfront treatment, and features of the relapse itself.

### 4.1. High Risk Clinical Features at Diagnosis

The clinical features present at diagnosis that have been found in multiple studies to be prognostic of poor outcome after relapse include primary tumor histologic subtype, site, stage and size. Alveolar histology has been identified as a risk factor both for relapse and for poor outcome post-relapse in IRS III-IV [8], the Italian RMS studies [4], the SIOP-MMT series [6], and the CWS studies [5]. In patients with localized disease treated on MMT-84, alveolar histology was associated with a greater risk of distant relapse [11]. Additionally, data from IRS III-IV demonstrated that for patients with tumors with alveolar histology, few other clinical features impacted post-relapse outcome, which was poor. In contrast, patients with tumors of botryoid histology had the best outcomes, superior to those classified with non-botryoid embryonal RMS (ERMS) tumors [8]. With the widespread use of molecular classification in RMS, the presence or absence of a *FOXO1* fusion has begun to supplant histological classification in risk stratification, as it has been increasingly recognized that tumors with these fusions are biologically and clinically distinct from those that lack them [18,19,20]. While nearly all tumors with non-alveolar histology lack a *FOXO1* fusion, about 20% of alveolar tumors are fusion negative, and behave more similarly to embryonal tumors [21,22]. A recent analysis identified *FOXO1* status as the most significant factor impacting outcomes among patients with localized RMS [3], suggesting that fusion status may be more predictive of outcome post-relapse than histology alone, although this has not been prospectively studied.

Results from IRS-I demonstrated that patients with non-alveolar primary tumors arising from the orbit and genitourinary (GU) system had the lowest risk of recurrence, while patients with perineal, gastrointestinal, pelvic or extremity tumors were at greatest risk for distant or multifocal recurrence [7]. In the early Italian RMS studies, patients with parameningeal (PM) or “other” primaries (defined as non-orbit, non-head and neck, non-extremity, non-GU) experienced the worst post-relapse outcomes, with 5-year OS of 0% and 19%, respectively, while those with primary tumors at orbital or non-bladder, non-prostate GU sites had the most favorable post-relapse outcome (5-year OS of 56% and 60%, respectively) [4]. The SIOP-MMT and CWS studies corroborated these findings, reporting that patients with primary tumors arising from unfavorable sites (PM, extremity, bladder/prostate, other) experienced increased risk of relapse and of post-relapse death [5,6]. A follow-up analysis of the IRS and SIOP-MMT data further refined the GU category, reporting that boys who are 10 years and older with non-metastatic paratesticular primary tumors had a significant risk of nodal relapse with poor post-relapse outcomes and should therefore undergo lymph node (LN) sampling at diagnosis [23,24]. Multivariate analysis performed on data from more recent studies, including RMS2005, have confirmed the earlier results, determining that unfavorable site is an independent adverse factor for post-relapse outcome in localized RMS [25].

Initial disease stage and tumor size have also been associated with post-relapse outcomes in multiple studies. In the IRS studies, patients with stage 4 disease at diagnosis were most likely to develop recurrent disease and for patients with non-metastatic ERMS at diagnosis, higher stage was associated with worse post-relapse survival [7,8]. A more recent prospective study for patients with relapsed RMS conducted by the Children’s Oncology Group (COG) similarly found that patients with lower stage ERMS upfront had superior post-relapse outcomes, compared to those with higher stages of upfront disease [26]. Data from the SIOP-MMT studies support these findings. Among patients without distant metastases at diagnosis, those with locoregional LN involvement (higher stage disease) experienced shorter post-relapse survival [6]. In addition, these studies found that patients who presented with tumors greater than 5 cm. had an increased risk of poor post-relapse outcome [6]. This finding was also reported for patients with ERMS and alveolar RMS (ARMS) tumors in the CWS studies [5,10].

### 4.2. Treatment-Related Risk Factors

Several factors related to the type of upfront treatment delivered have been found to be associated with post-relapse outcomes, including extent of surgical resection (clinical group), use of radiation therapy, and chemotherapy regimen. Clinical groups, as defined in the IRS studies, are as follows: group I—completely resected localized tumor; group II —localized tumor that is resected with positive margins or localized tumor with resected positive regional LNs; group III—localized tumor with gross residual disease post-biopsy or resection; group IV—tumor with distant metastasis [27]. Clinical group at initial diagnosis impacted post-relapse survival in the IRS III-IV studies, where patients with group I ARMS tumors experienced 5-year post-relapse survival of 40%, compared to 3% survival for those with group II-IV ARMS. For patients with ERMS, 5-year post-relapse survival was 52%, 20%, and 12% for those with group I, group II/III, and group IV disease, respectively [8]. These findings were confirmed in a COG study for patients with relapsed RMS, in which those who had been diagnosed with lower clinical group tumors experienced more favorable outcomes when treated with multiagent chemotherapy at relapse [26]. Clinical group has also been associated with post-relapse outcomes in infants, with patients with group I disease experiencing the best post-relapse outcomes and those with group III disease experiencing the poorest [14].

In addition to extent of initial surgical resection, the type and intensity of upfront therapy has been associated with differences in post-relapse outcome for patients with localized disease. In several studies, patients who had been treated with radiation as part of initial local control had poorer post-relapse outcomes than those who were treated with surgical control alone. The SIOP-MMT studies found prior radiation was independently associated with a shortened post-relapse survival with an odds ratio of 3.64 for patients who received upfront radiation compared to those who had not [6,11]. Similarly, the Italian studies found prior radiation to be an increased risk factor for post-relapse death, with 5-year post-relapse OS of 43% for those who had not received radiation versus 17% for those who had. However, in this analysis, radiation therapy was only significant in the univariate analysis, suggesting that other tumor characteristics that predisposed to the use of radiation, such as clinical group, site or size, may be confounding factors [4].

The number of agents used in upfront systemic chemotherapy has also been associated with post-relapse outcomes. Patients who were treated with an initial regimen with greater than two drugs experienced poorer outcomes following relapse in the SIOP-MMT series [6]. Data from IRS III-IV partially confirmed these findings, demonstrating a relationship between prior systemic chemotherapy and post-relapse outcome specifically in patients with orbital primary tumors. Those who received vincristine and actinomycin had a higher 5-year survival rate than those who had received more intensive regimens, although these findings may reflect confounding factors related to the choice of risk-directed therapy for these patients [8].

### 4.3. Features of Relapse

In addition to the features of initial disease presentation and treatment, several factors related to the nature of the relapse have been associated with post-relapse outcomes, including characteristics of relapse (i.e., local, regional or distant) and time to relapse. All major RMS consortia groups have reported on the significance of the type of relapse on subsequent outcome, finding that patients who recur at distant metastatic sites have the poorest outcomes, with reported survival rates ranging from 0–10% at 5 years after relapse [4,6,8,11,13]. A subgroup analysis of patients with localized ARMS similarly found that patients with “circumscribed” relapses, defined as a single disease focus located either at the primary site or elsewhere, had a 5-year post-relapse OS of 44%, compared to those with “widespread” relapses who had substantially poorer 5-year post-relapse OS of 2% [10]. In a prospective risk-stratified interventional relapse study conducted by the COG, the most favorable post-relapse outcomes were observed in those with botryoid or stage 1, group 1 ERMS tumors that had local/regional recurrences [26]. In addition to the location of the recurrent tumor, one report from the Italian RMS-96 and -2005 studies also identified that among patients with localized relapses, tumor size >5 cm at recurrence was the only significant factor associated with worse post-relapse outcome [25].

Time to relapse has also been found to correlate with subsequent outcome across numerous studies. Patients who recur or progress while on upfront therapy have the worst outcomes, with 5-year OS of between 2% and 8% in several reports [4,7]. For patients who complete upfront therapy in a complete response (CR) status, earlier relapses are associated with poorer survival rates [4,6]. In an analysis of the CWS studies, four-year post-relapse survival was 12%, 21%, and 41% for those who relapsed less than 6 months, between 6 and 12 months, and more than 12 months after completion of upfront therapy. Similar trends were noted when the study population was analyzed by histology, although OS rates were consistently worse for patients with ARMS tumors [12].

### 4.4. Risk Stratification Post-Relapse

Based on these datasets, several groups have developed criteria to define a group of patients with the highest likelihood of successful salvage therapy. For all sets of criteria, favorable characteristics include non-alveolar histology and local recurrence [4,6,8,26]. Additional favorable criteria specific to certain algorithms include botryoid histology [8,26], non-PM primary sites [4], group 1, stage 1 tumors [6,8,26], and recurrence after achievement of a CR [4]. Prior treatment regimens without radiation [4,6] and with just two (non-alkylator) chemotherapeutic agents [6,26] also stratify patients towards a more favorable outcome. Unfortunately, these criteria are met by a minority of relapsed patients [4,8,26].

## 5. Role of Early Detection of Relapse

After the completion of initial therapy for RMS, routine follow-up surveillance imaging is recommended by both the European and North American cooperative groups to monitor for recurrence [28,29]. Given the data on the prognostic value of time to relapse, the expectation is that earlier detection could improve post-relapse outcomes. Several studies have investigated the role of such imaging in the early detection of relapse. While systematic surveillance imaging has been shown to detect relapses earlier than presentation of clinical manifestations (both patient-reported and clinician-detected) [30], the majority of relapses are diagnosed based on clinical symptoms, namely pain or recognition of a new mass [28,29]. Data are inconclusive regarding whether surveillance imaging results in earlier detection of recurrence [28,30], however, in all studies, there was no impact on post-relapse response or OS [28,29,30]. Given the relatively small numbers of patients included in these studies, no subgroup analyses were performed. Therefore, it is possible that certain patient subgroups may benefit from earlier detection of relapse. In addition, most of the patients in these studies were followed with magnetic resonance imaging (MRI) or computed tomography (CT) scans. The role of fluorodeoxyglucose (FDG)-positron emission tomography (PET) on post-relapse outcomes has not yet been evaluated in the context of RMS surveillance and remains an open question. Furthermore, the potential utility of newer methods of relapse surveillance, such as detection of circulating tumor DNA or circulating tumor cells (CTCs), remains unknown.

## 6. Therapeutic Approach to Relapse

Although there is no universal standard regimen for relapsed RMS, several general principles can be applied to the approach for all patients experiencing first and subsequent relapses of RMS. These include biopsy confirmation of relapse diagnosis, assessment of post-relapse prognosis, determination of the feasibility of additional local control measures, and discussion of patient goals. These factors can then be used to consider which treatment options, a chemotherapy-based regimen, enrollment in a clinical trial, or palliative care, would be most appropriate going forward.

### 6.1. Assessment of Post-Relapse Prognosis

As described above, the collective North American and European clinical experience with patients with relapsed RMS defined a number of favorable features for post-relapse survival [4,6,8]. Based on the IRS III-IV studies, which defined favorable features as histologic subtype, disease group and stage [8], the COG conducted the first prospective clinical trial utilizing risk-based treatment for patients experiencing a first relapse of RMS [26,31]. Patients were defined as favorable-risk if they had local or locoregional recurrence of either botryoid RMS or stage 1 or group I ERMS at diagnosis and were not treated with cyclophosphamide in the upfront setting. This group (*n* = 14) was treated with a multiagent chemotherapy regimen of doxorubicin, cyclophosphamide, etoposide, ifosfamide (DCEI) for 32 weeks and experienced a 3-year failure-free survival (FFS) rate of 79%. Based on these data, it has been recommended that patients who meet these criteria at first relapse be treated with this type of regimen, before being offered other options, as they have the highest likelihood of achieving a long-term cure [26]. One consideration of these results is that some patients with botryoid RMS or stage 1 or group I ERMS are now receiving low-dose cyclophosphamide as part of an effort to reduce the length of upfront treatment, and it is unknown whether this prior exposure to cyclophosphamide may diminish the response to such a regimen in the relapse setting.

In the same study, patients who were defined as unfavorable-risk (*n* = 122) were randomized to a 6-week phase 2 window with one of two treatment schedules of irinotecan plus vincristine (VI), followed by an early disease assessment. Patients who achieved at least a partial response (PR) received an additional 44 weeks of VI plus DCEI (similar to the regimen used for the favorable-risk group); those who did not achieve a PR were switched to a 32-week regimen of DCEI plus a novel agent tirapazamine [26,31]. One-year FFS for patients treated on each of the arms containing irinotecan were 37% and 38% [31]. For those who did not achieve a PR after 6-weeks and continued on the DCEI plus tirapazamine arm, three-year FFS was 17% [26]. These results underscore the poor outcomes for the unfavorable-risk patients, despite the use of an intensified chemotherapy approach and suggest that even at first relapse, enrollment on clinical trials may be an appropriate choice.

### 6.2. Recent and Current Clinical Trials for Relapsed RMS

Despite both the advances in the understanding of RMS biology and the unmet need for new therapies, successful translational efforts to move new therapies into RMS-specific trials for patients have been limited. The reasons for this are numerous and may include a lack of sufficient preclinical data to justify human studies, regulatory barriers, and obstacles accessing new agents for study in the pediatric population [32]. In the last decade, there have been just five RMS-specific interventional trials opened for patients with relapsed RMS in North America and Europe, two of which have been completed and the other three which are currently open and enrolling.

The first, a large, randomized phase 2 study conducted by the COG (ARST0921), tested the use of a vinorelbine and cyclophosphamide backbone plus either the mammalian target of rapamycin (mTOR) inhibitor temsirolimus or the vascular endothelial growth factor (VEGF) inhibitor bevacizumab in patients with first relapse of RMS (NCT01222715). The selection of the backbone was based on clinical experience showing activity of vinorelbine in heavily pre-treated patients with RMS [33,34] as well as clinical data showing activity of vinorelbine and cyclophosphamide in a pilot study of relapsed sarcoma [35]. The novel agents were chosen based on robust preclinical data demonstrating evidence of the activity of VEGF [36] and mTOR inhibition [37,38] in RMS models and the availability of pediatric phase 1 data for both agents [39,40]. While there was no difference in response rate between the two arms, the patients who received temsirolimus experienced superior 6-month event-free survival (EFS) (69% vs. 55%) [41]. This finding provided the rationale to test temsirolimus in a randomized COG study for newly diagnosed intermediate-risk RMS patients (NCT02567435).

A second randomized phase 2 trial for patients with relapsed or refractory RMS was conducted by the European Soft Tissue Sarcoma Study Group (EpSSG) and tested the combination of vincristine and irinotecan with or without temozolomide (VI v. VIT) (NCT 01355445). Both preclinical [42,43,44] and clinical data [45,46,47] provide strong evidence of the activity of VI and VIT in RMS, including in the context of relapse [31,48,49,50]. In the EpSSG study, the overall response rate after 2 cycles was 44% for patients enrolled on the VIT arm, compared to 31% for those on the VI arm. In addition, progression-free survival (PFS) and OS were superior on the VIT arm, although when the analysis was restricted to exclude patients with refractory disease, the PFS and OS were similar [51]. The VI and VIT regimens will be further studied in the relapse setting in the recently activated EpSSG FaR-RMS study (NCT04625907), a complex multi-arm study enrolling both newly diagnosed and relapsed RMS patients, which will initially test the addition of the multikinase inhibitor regorafenib to the VI arm.

The final two trials specifically enrolling patients with relapsed RMS are smaller, single-arm studies testing novel combinations. The first is a phase 1/2 study for children and adults with relapsed or refractory ERMS or ARMS testing the combination of ganitumab, an antibody targeting the insulin-like growth factor receptor type 1 (IGF-1R) plus dasatinib, a multikinase inhibitor (NCT03041701). This study is based on early preclinical work [52] and clinical data [53] supporting the activity of inhibiting the IGF-1R pathway in RMS, and more recent work in animal models of RMS demonstrating that acquired resistance to single-agent IFG-1R therapy is mediated by bypass pathways that can be pharmacologically targeted for more durable disease control [54,55]. Given that the phase 2 study testing the addition of the IGF-1R antibody cixutumumab in an unselected population of stage IV RMS patients did not improve outcomes [56], the exploratory aims of the current clinical trial include biomarker studies to identify patients most likely to respond to this therapy. The other trial is a phase 1 study of the combination of vinorelbine and mocetinostat, a histone deacetylase (HDAC) inhibitor (NCT042991130). Several preclinical studies have demonstrated the importance of HDAC and the activity of HDAC inhibitors in models of fusion positive [57,58] and fusion negative RMS [59,60]. Both of these studies are open and enrolling.

A number of other early phase clinical trials that include patients with relapsed RMS among the eligible participants are also currently enrolling, but data have not yet been reported. These trials incorporate a variety of approaches, including testing novel agents (e.g., new cytotoxic drugs or targeted agents), mutation-based personalized treatments, immunotherapeutic approaches, and new local control modalities. A list of currently enrolling clinical trials that are specifically recruiting patients with relapsed RMS can be found in Table 1. Supporting published preclinical data in RMS models exists for some of these studies [61,62,63,64,65,66,67,68].

### 6.3. Chemotherapy Regimens for Relapsed RMS

In many cases, patients with RMS at first relapse are not enrolled on clinical trials and are treated with salvage chemotherapy, despite the suboptimal outcomes observed in the majority of patients. Based on the results of ARST0921, many oncologists have adopted vinorelbine/cyclophosphamide/temsirolimus as a standard regimen at first relapse in those patients with an unfavorable prognosis. However, with vinorelbine and cyclophosphamide increasingly being used both on- and off-clinical trials as maintenance therapy during primary treatment of RMS, other chemotherapy regimens may become more commonly employed in this setting. There have been no studies to directly compare the efficacy of these regimens, nor their optimal durations, and much of the available data on response rates come from single arm studies or single institution retrospective series. Table 2 lists the most commonly used regimens with available response data. Given the lack of robust data, the choice of salvage regimen for a given patient is generally based on which agents have not yet been used, the expected tolerability of potential toxicities, and patient and family preference. This becomes especially true for second and later relapses.

### 6.4. Other Systemic Approaches for Relapsed RMS

In addition to combination chemotherapy regimens, several other systemic therapies are sometimes pursued in the context of relapsed RMS. These are generally reserved for second or later relapse due to the lack of supportive efficacy data and include tyrosine kinase inhibitors, immunotherapeutics, and autologous or allogeneic stem cell transplants. For example, pazopanib, a multikinase inhibitor approved for use in soft tissue sarcoma, is occasionally used in the setting of relapsed RMS. Although no responses were observed in a small phase 2 cohort study conducted by the COG, case reports exist of patients with relapsed RMS responding to single-agent pazopanib [76]. Similarly, multiple studies have demonstrated that the use of high dose chemotherapy followed by autologous stem cell transplant lacks evidence of efficacy in relapsed RMS [77]. Allogeneic stem cell transplants have also not been effective in relapsed RMS [78]. Nonetheless, ongoing clinical trials continue to evaluate these approaches. Finally, immune checkpoint inhibitors, which have generated enthusiasm due to efficacy in other soft tissue sarcomas, have yet to demonstrate activity in relapsed RMS [79] but are occasionally used in this setting. Efforts to develop and evaluate new therapeutic approaches for RMS are ongoing (reviewed in [80,81]).

### 6.5. Local Control in Relapsed RMS

The approach to local control at the time of relapse depends on a number of factors including the nature of the prior local control, the feasibility of delivering additional local control, and the extent of the relapse. There is evidence that for patients with single site relapses, the type of local control delivered may meaningfully impact patient outcomes post-relapse. It is important to emphasize however, that in studies evaluating the role of local control, all patients received systemic chemotherapy as part of their treatments, underscoring the need for a multimodal approach that addresses the risks of both local and distant recurrence.

Since the earliest RMS studies, there have been reports on the potential benefits of surgery for patients with localized recurrences. In relapsed patients from IRS-I, those who underwent complete surgical excision of the recurrent tumor experienced superior outcomes [7]. Data from the Italian studies RMS-88 and -96 similarly demonstrated that for patients who presented with local recurrences, OS was 54% for patients who had undergone surgical excision, compared to 25% for those who had not. In this series, heroic radical surgery did not yield better outcomes than conservative surgery [82]. In a more recent retrospective, single-institution study of patients with recurrent RMS (all stages), resection improved 5-year survival from 8% to 37%, although patients who did not undergo resections typically had multifocal bony disease that was not amenable to surgery, thus introducing a selection bias. Notably, in the surgical patients, the type of surgery (aggressive v. conservative) did not impact outcome [83].

The addition of radiation therapy to surgical management of recurrence has been shown to be beneficial in multiple studies. In RMS-88 and -96, patients who underwent surgical intervention plus radiation experienced superior OS (61%) compared to those who underwent surgical intervention alone (42%) [82]. In the CWS-91 relapse study, patients who were treated with radiation as part of their salvage therapy experienced superior 3-year post-relapse EFS (46% v. 10%) [84]. Data from the CWS series reporting on patients with locally relapsed ARMS who had circumscribed relapses, defined as a single disease focus located either at the primary site or elsewhere, patients who received adequate local control, defined as either complete resection or gross resection plus radiation, experienced 5-year post-relapse survival of 54%, which was superior to those with circumscribed relapses who did not receive adequate local control (5-year post-relapse survival of 27%) or those with widespread relapses (5-year post-relapse survival of 9%). In multivariate analysis, adequate local control was an independent prognostic factor for this group of patients [10].

For patients with localized recurrence who are unable to undergo surgical intervention at relapse, there is some evidence that radiation alone may also be beneficial [82,85]. For patients who lack an option for conventional re-irradiation at recurrence due to prior radiotherapy, a common issue for patients with primary tumors arising in the head and neck sites, site-specific mixed local control approaches, for example, the use of salvage AMORE (Ablative surgery, MOuld technique brachytherapy and surgical REconstruction) treatment, may be feasible and have been associated with promising results [86].

## 7. Conclusions

In summary, relapse remains a therapeutic challenge that affects a substantial proportion of patients with RMS. While historical and more recent clinical trials have provided much information regarding the factors that impact likelihood of relapse and prognosis post-relapse, there have been fewer advances that have resulted in the successful treatment of patients after relapse. For patients with favorable features at first relapse, salvage multiagent chemotherapy is an appropriate approach. However, for the vast majority of patients with unfavorable features at first relapse, strong consideration should be given to enrollment on clinical trials. Future clinical trials for relapsed RMS will likely be directed to subgroups that have specific molecular markers that can be therapeutically targeted.

## Figures and Tables

**Table 1 jcm-10-00804-t001:** Active clinical trials recruiting patients with relapsed RMS.

Intervention	Class	Study Population	NCT	Phase
Vincristine/Irinotecan +/−Temozolomide (FaR-RMS)	Cytotoxic	RMS	04625907	2
Ganitumab + Dasatinib	IGF-1R antibody + multikinase inhibitor	RMS	03041701	1/2
Vinorelbine + Mocetinostat	HDAC inhibitor + cytotoxic	RMS	04299113	1
Prexasertib + Irinotecan	CHEK2 inhibitor + cytotoxic	RMS, DSRCT	04095221	1/2
Olaparib + Temozolomide	PARP inhibitor + cytotoxic	RMS, EWS	01858168	1
Eribulin mesylate	Cytotoxic	RMS, NRSTS, EWS	03441360	2
Eribulin mesylate + Irinotecan	Cytotoxic	Solid tumors	03245450	1/2
Nab-paclitaxel + Gemcitabine	Cytotoxic	RMS, NRSTS, OST, EWS	02945800	2
Nab-paclitaxel + Gemcitabine	Cytotoxic	Pediatric solid tumors	03507491	1
Regorafenib	Multikinase inhibitor	RMS, EWS, OST, LPS, MCS	02048371	2
Abemaciclib	CDK4/6 inhibitor	Pediatric solid tumors	02644460	1
Abemaciclib + Irinotecan or Irinotecan/Temozolomide	CDK4/6 inhibitor + cytotoxic	Pediatric solid tumors	04238819	1
Copanlisib	PI3K inhibitor	Pediatric solid tumors	03458728	1/2
Vorinostat + chemotherapy	HDAC inhibitor + cytotoxic	Solid tumors	04308330	1
Sirolimus + metronomic chemotherapy	mTOR inhibitor + cytotoxic	Pediatric solid tumors	02574728	2
High-dose alkylator chemotherapy + autologous transplant	Cytotoxic + cellular rescue	Solid tumors	01505569	2
Pediatric MATCH	Personalized	Pediatric solid tumors	Multiple	2
ESMART	Personalized	Pediatric malignancies	02813135	1/2
B7H3 CAR T Cells	Immunotherapy	Pediatric solid tumors	04483778	1
GD2 CAR T Cells	Immunotherapy	Pediatric solid tumors	03635632	1
EGFR CAR T Cells	Immunotherapy	Pediatric solid tumors	03618381	1
Allogeneic HSCT	Transplant	Pediatric solid tumors	04530487	2
Haploidentical HSCT + Zometa	Transplant	Pediatric malignancies	02508038	1
Reduced intensity haploidentical HSCT	Transplant	Solid tumors	01804634	2
Haploidentical NK cells	Transplant	RMS, EWS	02409576	1/2
Universal donor NK cells + ALT803	Transplant	Malignancies	02890758	1
High intensity focused ultrasound (HIFU)	Local control	Pediatric solid tumors	02076906	1
HIFU + thermosensitive liposomal doxorubicin	Local control	Pediatric solid tumors	02536183	1
CLR-131	Local control	Pediatric solid tumors	03478462	1

CAR: chimeric antigen receptor; HSCT: hematopoietic stem cell transplant; NK: natural killer; IGF-1R: insulin-like growth factor receptor, type 1; HDAC: histone deacetylase; CHEK2: checkpoint kinase 2; PARP: poly-ADP ribose polymerase; CDK: cyclin dependent kinase; PI3K: phosphoinositide 3 kinase; mTOR: mammalian target of rapamycin; RMS: rhabdomyosarcoma; DSRCT: desmoplastic small round cell tumor; EWS: Ewing sarcoma; NRSTS: non-rhabdomyosarcoma soft tissue sarcoma; OST: osteosarcoma; LPS: liposarcoma, MCS: mesenchymal chondrosarcoma.

**Table 2 jcm-10-00804-t002:** Combination chemotherapy regimens commonly used for patients with relapsed RMS.

Regimen	Outcome Data	Study Design; Number of Patients
Doxorubicin, cyclophosphamide, ifosfamide, etoposide	Favorable risk: 3-year FFS 79% Unfavorable risk: 3-year FFS 17%	RCT [26] Favorable: *n* = 14 Unfavorable: *n* = 122
Vinorelbine, cyclophosphamide, temsirolimus	6-month EFS: 69% RR: 47%	RCT (v. vinorelbine, cyclophosphamide, bevacizumab) [41] *n* = 87
Vinorelbine, oral cyclophosphamide	Median survival: 9 months RR: 36%	Phase 2 single arm [69] *n* = 50
Vincristine, irinotecan, temozolomide	3-month PFS: 23% DCR: 27%	Retrospective analysis [49] *n* = 19
	RR: 43%	Retrospective analysis [70] *n* = 7
	RR: 25%	Retrospective analysis (ARMS) [48] *n* = 4
Vincristine, irinotecan	1-year FFS: 37–38% RR: 26–37%	Randomized phase 2 window (two schedules) [31] *n* = 92
Cyclophosphamide, topotecan	Median time to progression: 2 months RR: 67%	Phase 2 single arm [71] *n* = 15
	RR: 67%	Retrospective analysis [72] *n* = 6
Ifosfamide, carboplatin, etoposide	2-year OS: 26% 2-year OS for ERMS: 46% RR: 67%	Retrospective analysis [73] *n* = 27
Topotecan, carboplatin	5-year PFS: 17% RR: 28%	Single arm [74] *n* = 38
Gemcitabine, docetaxel	RR: 40%	Retrospective single institution [75] *n* = 5

FFS: failure-free survival; EFS: event-free survival; DCR: disease control rate; RR: response rate; PFS: progression-free survival; RCT: randomized clinical trial.

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
