# Peer review of "Relapsed Rhabdomyosarcoma"

_jcm, 2021, doi:10.3390/jcm10040804_

Round 1
Reviewer 1 Report
Dear Authors,
the paper is well written and reportes a large view in relapsed rhabdomyosarcoma.
To my, even if it is mainly focused in COG expertise, this review resumed all emerged consideration across the larger groups worldwide involved in the protocols development on RMS, with a well detailed excursus on treatment options and prognostic factors, able to give suggestion on how to approach relapsed RMS.
By this consideration, on the other hand, this review did not bring any new concepts or discuss about new approaches such as CAR-T cell therapies or immunotherapies. It remains superficial compared to what we have already done in relapsed RMS.
To be intresting maybe, some paragraphs might be omitted or groupped, inserting of a deteiled discussion on these innovative treatments approach and molecular targets for relapsed RMS in which prognosis remeined the same in the last 20 yrs.
I suggest to apport moderate revisions in the structure and contents of this review.
Author Response
Please see the attachment.
Reviewer 1:
Dear Authors,
the paper is well written and reportes a large view in relapsed rhabdomyosarcoma.
We thank the reviewer for this comment.
To my, even if it is mainly focused in COG expertise, this review resumed all emerged consideration across the larger groups worldwide involved in the protocols development on RMS, with a well detailed excursus on treatment options and prognostic factors, able to give suggestion on how to approach relapsed RMS.
We thank the reviewer for acknowledging the broad scope of this review.
By this consideration, on the other hand, this review did not bring any new concepts or discuss about new approaches such as CAR-T cell therapies or immunotherapies. It remains superficial compared to what we have already done in relapsed RMS.
We have incorporated language about a number of emerging therapeutic approaches for relapsed RMS in section 6.2 and Table 1. This section references a number of ongoing clinical trials, including three CAR T-cell trials and five transplant/immunotherapy trials. As there are no published data on these studies, a more extensive discussion of these trials cannot be included in this review. We have revised the text to reflect this (page 7, lines 345-346).
We have chosen to focus the extended discussion in this section on active clinical trials that are specific to patients with relapsed RMS (as opposed to trials that include patients with RMS among a larger group of eligible solid tumors) as there are more robust biological and preclinical rationale and data for such studies. That said, in the interest of inclusion, Table 1 does include studies that are not RMS-specific, though all contain specific eligibility language for RMS. Previously-reported negative clinical trials using immunotherapy for relapsed RMS and preclinical studies on immunotherapy in RMS that may provide data for future therapies are beyond the scope of this review.
Section 6.4 also addresses the use of immunotherapeutics in relapsed RMS, specifically the use of allogeneic transplants and checkpoint inhibitors, with references to the published literature. These have been included since they are currently in clinical use by some providers and thus, we feel the data are relevant to the readership of this article.
We have not been able to find additional publications demonstrating the effectiveness of immunotherapy for relapsed rhabdomyosarcoma in the clinical setting.
To be intresting maybe, some paragraphs might be omitted or groupped, inserting of a deteiled discussion on these innovative treatments approach and molecular targets for relapsed RMS in which prognosis remeined the same in the last 20 yrs.
In addition to the inclusion of immunotherapy studies in Table 1, we have also included ten studies that address molecular targets for relapsed RMS. As described above, there is not published clinical data that we are able to describe, as these trials are ongoing. A broader discussion of emerging RMS therapies is outside the scope of this manuscript and multiple recent reviews have addressed this topic (for example: Chen et al, Frontiers in Oncology, 2019; Miwa et al, Cancers, 2020). These are now referenced in the text (page 11, lines 390-391).
I suggest to apport moderate revisions in the structure and contents of this review.

Reviewer 2 Report
an excellent review of the literature a few minor comments
as reference is made to the IRS studies and clinical groups, whether a summary table of these studies may be provided for the reader
4.2 treatment related risk factors
the authors comment on radiation as an independent poor prognostic factor for relapse; can they confirm this is not a confounding factor associated with site, inoperability and size?
do the authors have any comment on age as a risk factor for relapse? particularly the adult cohort,
do the authors have any comment on delivery of treatment and maintaining dose intensity and also the role of maintenance chemo to influence relapse
4.4 risk stratification
any evidence age is a risk factor?
5 role of early detection of relapse
what imaging would the authors recommend? is their a role for PETCT imaging?
Author Response
Please see the attachment.
Reviewer 2:
an excellent review of the literature a few minor comments
We thank the reviewer for this comment.
as reference is made to the IRS studies and clinical groups, whether a summary table of these studies may be provided for the reader
We have added in a description of the IRS clinical groups at the first mention of clinical group (page 2, lines 86-87) and more fully on page 4, lines 160-164.
4.2 treatment related risk factors
the authors comment on radiation as an independent poor prognostic factor for relapse; can they confirm this is not a confounding factor associated with site, inoperability and size?
These studies look at radiation therapy as a poor prognostic factor for post-relapse survival, not relapse itself. That said, in the SIOP-MMT studies, prior radiation did emerge as a significant factor for post-relapse survival in the multivariate analysis, which included site, clinical group, and size. However, in the Italian study, XRT was significant in univariate, but not multivariate analysis, which suggests that other tumor characteristics could be confounders. We have edited the text describing the Italian study to better reflect this (page 4, lines 184-187).
do the authors have any comment on age as a risk factor for relapse? particularly the adult cohort,
Data evaluating relapse in infants suggests that very young patients suffer higher local relapse rates however, this is generally attributable to differences in therapeutic choices. This is described on page 2, lines 96-98 and page 3, line 100.
Age as a prognostic factor post-relapse has not been found to be significant in any of the European or IRS studies with published data. However, these studies have primarily enrolled patients 18 and younger, so direct data comparison with adult patients is lacking. While we are aware that overall outcomes for adults with RMS are worse than those for children, the one study that has analyzed the potential reasons for this was a single institution retrospective analysis which concluded that for adult patients whose treatment adhered to pediatric treatment guidelines, outcomes were similar to those reported in pediatric series (Ferrari et al, Cancer, 2003). However, there are not published studies analyzing any other potential factors relevant for post-relapse outcomes in adult patients. The more recent inclusion of adult patients on consortia trials will hopefully address some of these knowledge gaps.
do the authors have any comment on delivery of treatment and maintaining dose intensity and also the role of maintenance chemo to influence relapse
These are open questions in the field that have yet to be answered. While studies are suggestive that dose intensity, particularly of cyclophosphamide, may impact the rate of relapse in some RMS subsets, there is not data comparing post-relapse outcomes of patients who received different dose intensity in their upfront regimens.
Similarly, while the data from RMS2005 indicate a benefit from maintenance therapy for some RMS patients, the benefit was mainly in OS and not in EFS (Bisogno et al, Lancet, 2019). This suggests that maintenance therapy in the context of RMS2005 induction therapy may affect relapse pattern, although post-relapse outcomes for patients who have received maintenance compared to those who have not, have not been reported. In addition, in other studies, more intensive induction chemotherapy without maintenance had similar outcomes to those on the RMS 2005 maintenance arm, although the patient groups studied were not identical. Future studies will be necessary to better answer these questions.
4.4 risk stratification
any evidence age is a risk factor?
Based on the available data, there is not evidence that age is a risk-factor for post-relapse survival in non-infant, pediatric and adolescent patients.
5 role of early detection of relapse
what imaging would the authors recommend? is their a role for PETCT imaging?
There are no studies in the literature indicating that surveillance with PET/CT will impact outcome. However, of the few studies in RMS that have looked at the role of surveillance imaging, the minority of patients were followed with PET/CT. Without any data, it is not possible to recommend for or against PET/CT as a modality to detect relapse, but we have revised the text to reflect that this remains an open question (page 5, lines 249-251).

Round 2
Reviewer 1 Report
Dear Authors,
thank you for your comments, the new version is endorsed for publication and respect my considerations.
Thank you for the effort to implement immunotherapy, to note at my institution is ongoing the NCT03373097 trial with enrolling also GD2 in pediatric sarcoma.